# Histone H3K36me2 and H3K36me3 form a chromatin platform essential for DNMT3A-dependent DNA methylation in mouse oocytes

Seiichi Yano [1,2], Takashi Ishiuchi [1,3] ✉, Shusaku Abe[1], Satoshi H. Namekawa [4], Gang Huang[5], Yoshihiro Ogawa[2] & Hiroyuki Sasaki [1] ✉

Establishment of the DNA methylation landscape of mammalian oocytes, mediated by the DNMT3A-DNMT3L complex, is crucial for reproduction and development. In mouse oocytes, high levels of DNA methylation occur exclusively in the transcriptionally active regions, with moderate to low levels of methylation in other regions. Histone H3K36me3 mediates the high levels of methylation in the transcribed regions; however, it is unknown which histone mark guides the methylation in the other regions. Here, we show that, in mouse oocytes, H3K36me2 is highly enriched in the X chromosome and is broadly distributed across all autosomes. Upon H3K36me2 depletion, DNA methylation in moderately methylated regions is selectively affected, and a methylation pattern unique to the X chromosome is switched to an autosome-like pattern. Furthermore, we find that simultaneous depletion of H3K36me2 and H3K36me3 results in global hypomethylation, comparable to that of DNMT3A depletion. Therefore, the two histone marks jointly provide the chromatin platform essential for guiding DNMT3A-dependent DNA methylation in mouse oocytes.

The unique DNA methylation landscape of mammalian oocytes is crucial for reproduction and development, and its formation is mediated by the protein complex composed of the de novo methyltransferase DNMT3A and its cofactor DNMT3L[1–5]. In mouse oocytes, the transcriptionally active regions are highly DNA methylated, while other regions show moderate to low levels of methylation[6–9]. It has been suggested that recognition of specific histone marks by the DNMT3A-DNMT3L complex is important for the region-selective de novo DNA methylation[5,8–11]. However, depletion of the histone-modifying enzyme for H3K36me3,

H3K4me3, or H3K9me2 causes only partial loss or redistribution of DNA methylation[12–14]. For example, a loss of H3K36me3 decreases the level of DNA methylation in transcriptionally active regions, but the level of methylation in the other regions is either unchanged or even increased[14]. These findings suggest that a combination of specific histone marks or an additional mark may be operative.

Recently, it was reported that H3K36me2 contributes to de novo DNA methylation in several cell types[15–17]. This is in line with the fact that DNMT3A has a Pro-Trp-Trp-Pro (PWWP) domain,

[1]Division of Epigenomics and Development, Medical Institute of Bioregulation, Kyushu University, Fukuoka, Japan. [2]Department of Medicine and Bioregulatory Science, Graduate School of Medical Sciences, Kyushu University, Fukuoka, Japan. [3]Faculty of Life and Environmental Sciences, University of Yamanashi, Yamanashi, Japan. [4]Department of Microbiology & Molecular Genetics, University of California Davis, Davis, CA, USA. [5]Department of Cell Systems & Anatomy and Department of Pathology & Laboratory Medicine, UT Health San Antonio, Joe R. and Teresa Lozano Long School of Medicine, San Antonio, TX, USA. ✉e-mail: tishiuchi@yamanashi.ac.jp; hsasaki@bioreg.kyushu-u.ac.jp

which recognizes both H3K36me2 and H3K36me3[10]. The crosstalk between H3K36me2 and DNA methylation is also suggested in human diseases; mutations in *DNMT3A* and *NSD1* (encoding a histone methyltransferase for H3K36me2) cause phenotypically similar overgrowth disorders called Tatton–Brown–Rahman syndrome[18] and Sotos syndrome[16], respectively. In addition, a subset of head and neck squamous cell carcinoma harboring *NSD1* mutations show DNA hypomethylation[17]. In this study, we examine the possible involvement of this histone mark in de novo DNA methylation in oocytes and find that H3K36me2 is selectively required for DNA methylation in moderately methylated regions. Furthermore, we show that simultaneous loss of H3K36me2 and H3K36me3 causes global DNA hypomethylation comparable to that in *Dnmt3a* knockout oocytes. Thus, our results reveal that H3K36me2 and H3K36me3 together constitute the primary platform for the establishment of the DNA methylation landscape of mammalian oocytes.

## Results

### H3K36me2-enriched regions are associated with moderate levels of CG methylation in oocytes

We first explored the distribution of H3K36me2 using immunofluorescence. We observed broad H3K36me2 staining in the germinal vesicle and strong H3K36me2 signals in a region near the nuclear envelope of fully grown oocytes (FGOs) (Fig. 1a and Supplementary Fig. 1a), while the H3K36me3 signals colocalized with nucleic acid staining by 4′6-diamino-2-phenylindole (DAPI) (Supplementary Fig. 1b). We then examined published Lamin B1-DamID data[19] (Supplementary Table 1); the X chromosome was specifically associated with the nuclear lamina of FGOs (Supplementary Fig. 1c, d), suggesting that the strong H3K36me2 signals near the nuclear envelope could represent the X chromosome.

To elucidate the H3K36me2 distribution in more detail, we performed ultra-low-input native chromatin immunoprecipitation sequencing (ULI-NChIP-seq)[20] (Supplementary Table 2). H3K36me2

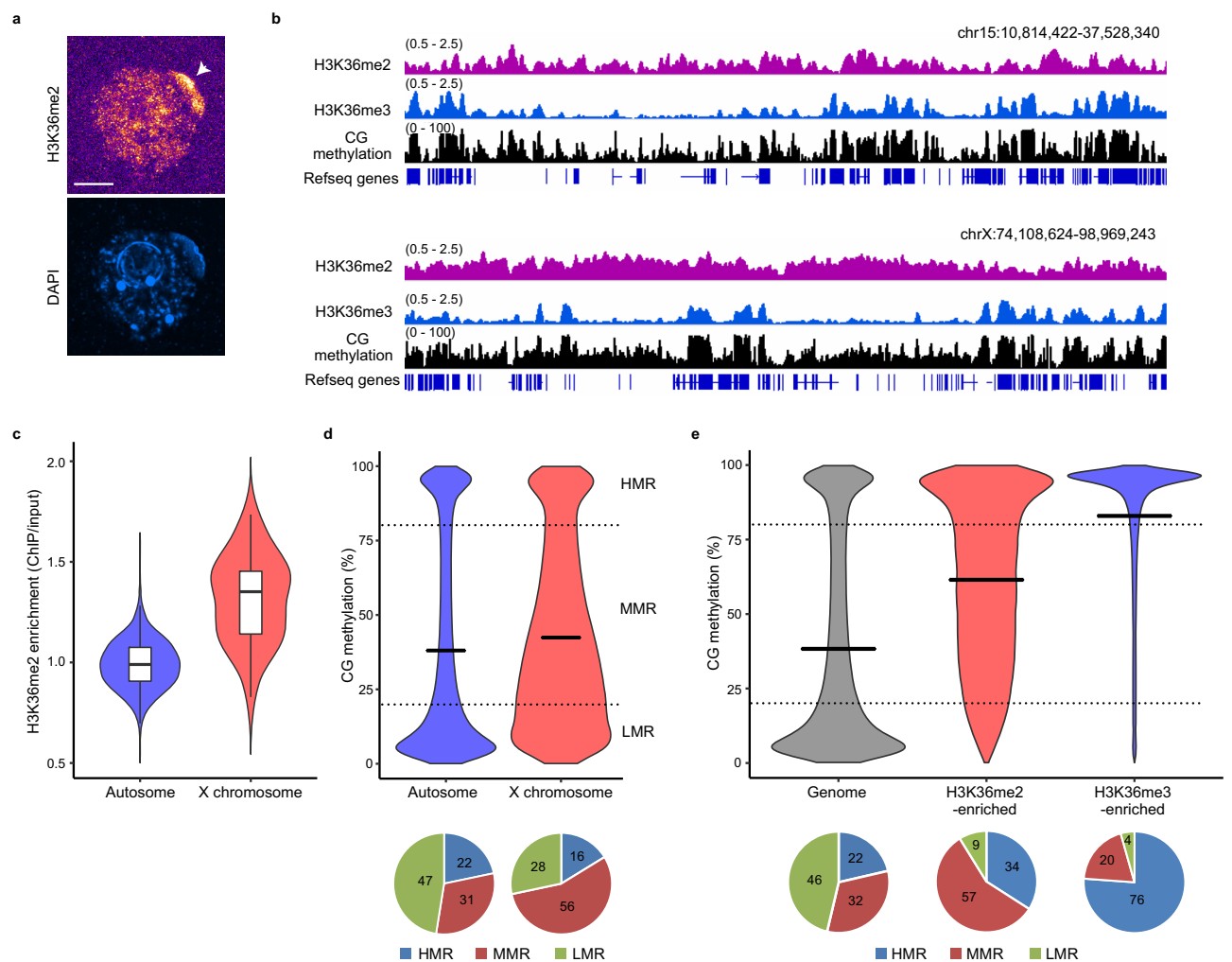

**Fig. 1 | H3K36me2-enriched regions are associated with moderate levels of CG methylation in FGOs. a** Representative images of germinal vesicles of FGOs immunostained for H3K36me2 (*n* = 14, from two independent experiments). Other representative images are also shown in Supplementary Fig. 1a. An arrowhead indicates relatively strong H3K36me2 signals near the nuclear envelope. The maximum intensity projection images are shown. DAPI 4′,6-diamidino-2-phenylindol. Scale bar, 10 μm. **b** Genome browser snapshots showing H3K36me2 and H3K36me3 enrichment and CG methylation in FGOs. The upper and lower panels show the representative regions of chromosomes 15 and X, respectively. H3K36me2 and H3K36me3 enrichment are indicated by ChIP/input. **c** Violin plots showing H3K36me2 enrichment in 5 Mb bins in autosomes (*n* = 502) and X

chromosome (*n* = 35). Boxplots show median value and 25–75th percentiles, and whiskers show 1.5× interquartile range from the box. Source data are provided as a Source Data file. **d** Violin plots showing CG methylation levels of 10 kb bins in autosomes (*n* = 238,084) and the X chromosome (*n* = 15,494). The methylation level was determined in 10 kb bins. Horizontal bars indicate mean values. Pie charts show percentages of 10 kb bins categorized as HMRs, MMRs, and LMRs. **e** Violin plots showing CG methylation levels of 10 kb bins across the genome (*n* = 253,578), H3K36me2-enriched regions (ChIP/input ≥ 1.4, *n* = 30,234), and H3K36me3-enriched regions (ChIP/input ≥ 1.5, *n* = 28,655). Horizontal bars indicate mean values. Pie charts show percentages of 10 kb bins categorized as HMRs, MMRs, and LMRs.

was distributed in broad domains covering both genic and intergenic regions in FGOs, albeit with local changes, as observed in cultured cells and mouse tissues[21–25] (Fig. 1b). (See later for more detailed characterization of the distribution in relation to the transcriptional activity and other histone marks.) However, H3K36me2 enrichment occurred in larger regions in the X chromosome (Fig. 1b, c and Supplementary Fig. 1e), which in total occupied 45% of this chromosome (compared to 10% in the autosomes; 10 kb bins, ChIP/input ≥ 1.4); such biased enrichment was not observed for H3K36me3 (Supplementary Fig. 1f, g) (Supplementary Table 2).

DNA methylation mainly occurs at the cytosine of CG dinucleotides in almost all cell types, including oocytes, although non-CG methylation is also prevalent in FGOs[7]. To explore the possible link between H3K36me2 and CG methylation, we generated a whole-genome bisulfite sequencing (WGBS) map of FGOs (Supplementary Table 3). After confirming the reproducibility in biological replicates (Supplementary Note), we combined the data and categorized 10 kb regions (bins) into highly CG methylated (≥80%), moderately methylated (20–80%), and hypomethylated regions (<20%) (designated as HMRs, MMRs, and LMRs, respectively). In all chromosomes, the regional CG methylation exhibited a bimodal distribution with a preference for HMR and LMR; however, MMR was more frequent in the X chromosome (Fig. 1b, d and Supplementary Fig. 1h). Therefore, H3K36me2 could be linked to a moderate level of CG methylation. Indeed, the H3K36me2-enriched regions were more frequently associated with MMRs, while H3K36me3-enriched regions with HMRs (Fig. 1e and Supplementary Fig. 2a, b), suggesting independent functioning of the two marks in CG methylation regulation.

### Loss of H3K36me2 causes CG hypomethylation in MMRs of oocytes

Histone H3.3 with a K36M substitution (H3.3K36M) was identified as an onco-histone that affects the global H3K36 methylation level through its inhibitory effect on histone H3K36 methyltransferases[17,24–26]. While several studies reported that the mutation affects both H3K36me2 and H3K36me3[17,24,25,27,28], we and others found that, in mouse tissues, H3.3K36M causes specific reduction of H3K36me2, possibly through its preferential effect on NSD1 and NSD2[21,22], which catalyze H3K36me2 deposition. To determine the importance of H3K36me2 in shaping the CG methylation landscape of oocytes, we used conditional knock-in mice that we reported previously[22]: this mouse line expressed HA-tagged H3.3K36M from the *H3f3b* locus in a Cre-dependent manner (Supplementary Fig. 3a). (No detrimental effect of the HA-tag itself was confirmed by expressing an HA-tagged wild-type H3.3 protein[22].) We collected FGOs from control (*H3f3b*[K36M-flox/+]) and *K36M* (*H3f3b*[K36M-flox/+]; *Gdf9*-Cre) mice (the *Gdf9* promoter drives Cre in an oocyte-specific manner[29]; Supplementary Fig. 3b) and performed immunofluorescence for H3K36me2 and H3K36me3. H3.3K36M caused a global loss of H3K36me2 (Fig. 2a), with virtually no effect on H3K36me3 in FGOs (Supplementary Fig. 3c). ULI-NChIP-seq with a spike-in control (Supplementary Table 2) showed that the level of H3K36me2, but not H3K36me3, was substantially decreased (Fig. 2b, c and Supplementary Fig. 3d, e).

We next generated WGBS maps of *K36M* and control FGOs (Supplementary Table 3). Unlike H3K36me3 depletion[14], the H3K36me2 depletion did not affect the imprinting control regions (Supplementary Fig. 3f). However, the level of global CG methylation was lower in *K36M* FGOs (31.0%) than in control FGOs (36.4%), which was mainly attributable to decreased CG methylation of the MMRs (Fig. 2d, e). The X chromosome was the most severely affected (Fig. 2f). The unique CG methylation pattern observed in the X chromosome was switched to an autosome-like pattern (Fig. 2e). The CG methylation level was decreased in H3K36me2-enriched regions (Fig. 2d and Supplementary Fig. 4a) but not in H3K36me3-enriched regions (Supplementary Fig. 4b). The extent of H3K36me2 enrichment correlated well with the

decrease in CG methylation (Fig. 2g). Taken together, these results indicate that H3K36me2 adjusts CG methylation levels at MMRs in FGOs.

### Loss of H3K36me2 has a limited effect on the transcriptomes of oocytes and two-cell embryos

Since H3K36me2 inhibits deposition of H3K27me3[23,30,31], which has a role in suppressing genes in oocytes and preimplantation embryos[32], we examined the distribution of H3K27me3 in *K36M* FGOs using CUT&RUN[33,34] (Supplementary Table 2). We grouped 10 kb genomic bins into five clusters based on H3K36me2 and H3K36me3 enrichment statuses in the control FGOs. H3K36me2/3 and H3K27me3 showed a clear reciprocal enrichment pattern in the control FGOs; H3K36me2 depletion apparently caused little change in H3K27me3 enrichment in all clusters (Fig. 3a). When we focused on the X chromosome, a slight increase in H3K27me3 was observed in regions normally enriched for H3K36me2 (clusters 2 and 4) (Supplementary Fig. 5a). When we selected 2407 genes with reduced H3K36me2 enrichment in the *K36M* FGOs (Supplementary Data 1), they were associated with H3K27me3 gain, irrespective of the chromosome (Fig. 3b, c). The CG methylation level decreased at these loci (Supplementary Fig. 5b).

We then performed transcriptome analysis (Supplementary Table 2) and found that, in control FGOs, H3K36me2-enriched regions (genes) were transcribed at low levels, while H3K36me3-enriched regions (genes) were transcribed at much higher levels (Supplementary Fig. 5c) (Supplementary Data 2, 3). Furthermore, only 54 genes were differentially expressed (FDR < 0.05) between *K36M* and control FGOs and only one gene between *K36M* oocyte-derived and control oocyte-derived two-cell embryos (Fig. 3d and Supplementary Data 4). The transcripts from the maternal and paternal alleles were analyzed individually in two-cell embryos; irrespective of the allelic origin, none was expressed differentially between *K36M* oocyte-derived and control embryos (Supplementary Fig. 5d). Genes that lost H3K36me2 (and hence CG methylation) and gained H3K27me3 in *K36M* FGOs were only weakly expressed in both control FGOs and two-cell embryos (Fig. 3e); this likely explains the little impact of observed histone mark changes on gene expression. These findings show that a loss of H3K36me2 affects H3K27me3 and CG methylation, especially in the X chromosome, but has little impact on the transcriptome in oocytes and two-cell embryos.

### Maternal H3.3K36M leads to embryonic lethality around implantation

To understand the impact of the *K36M* mutation in oocytes on embryonic development, we crossed *K36M* females with wild-type males, which should generate offspring of two genotypes *H3f3b*[+/+] and *H3f3b*[K36M/+] (see the experimental scheme in Supplementary Fig. 6a). No pup was obtained from these females, and no embryo was recovered even at E6.5 (Supplementary Fig. 6b, c). We then performed in vitro fertilization; the embryos developed normally until the blastocyst stage (Supplementary Fig. 6d), suggesting that the lethality arises around implantation. Then, immunostaining revealed that, while H3.3K36M protein was detected in *H3f3b*[K36M/+] blastocysts, it was undetectable in *H3f3b*[+/+] blastocysts (Supplementary Fig. 6a), indicating that maternal H3.3K36M was already degraded at this stage. However, the H3K36me2 signal was not yet fully recovered in not only *H3f3b*[K36M/+] but also *H3f3b*[+/+] blastocysts (Supplementary Fig. 6a). These data indicate that maternal H3.3K36M expression causes H3K36me2 loss persisting to the blastocyst stage as well as embryonic lethality around implantation.

Although the above study demonstrated the detrimental effect of maternal H3.3K36M, early *H3f3b*[+/+] embryos could have a carry-over of the mutated protein, albeit undetectable in blastocysts, which could contribute to the phenotype. To examine the impact of zygotic H3.3K36M on early development, we expressed the mutated protein

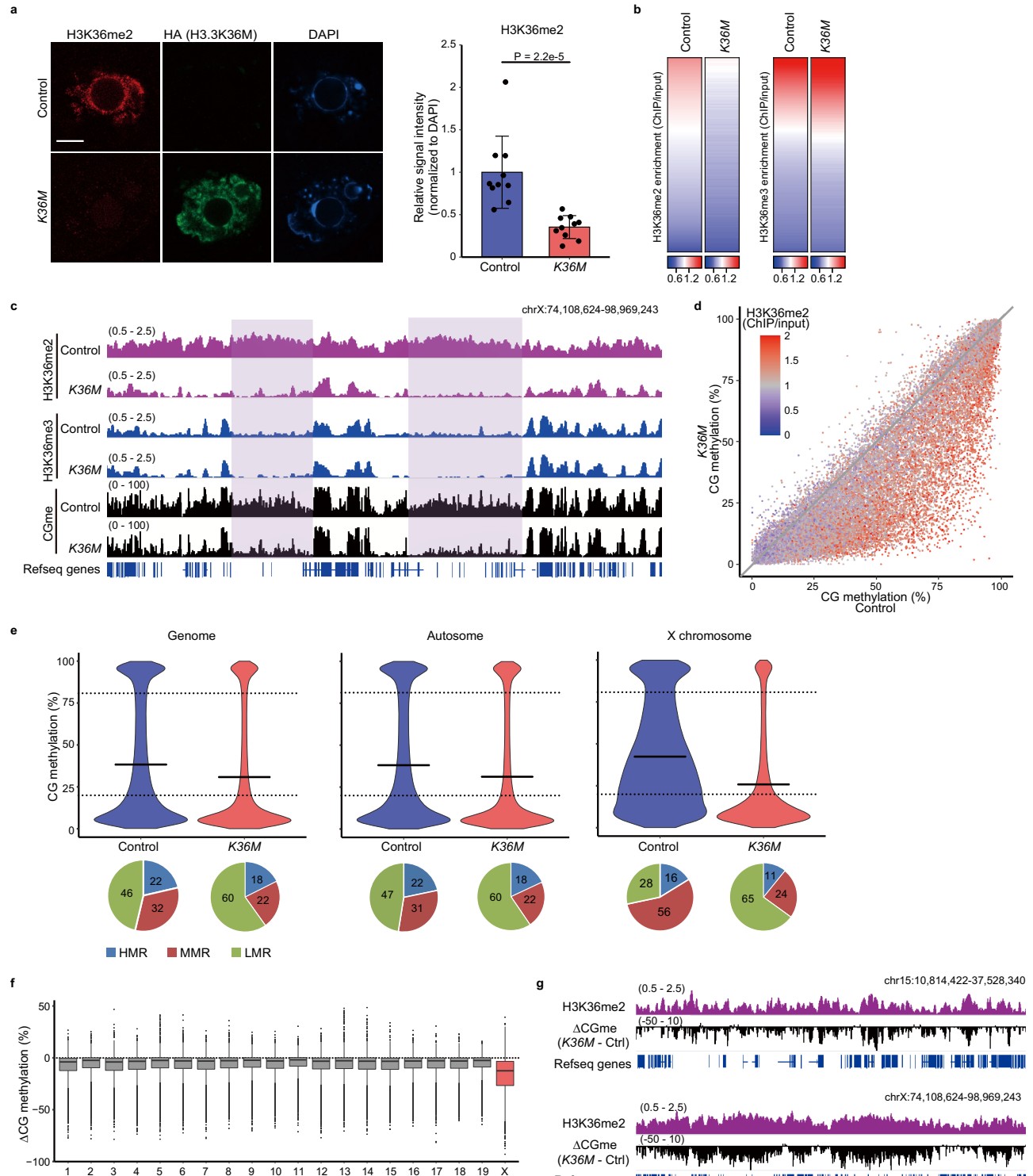

only after the zygote stage by fertilizing *Gdf9-Cre* oocytes with *H3f3b*^K36M-flox/K36M-flox sperm in vitro, expecting the zygotic action of maternal Cre (Supplementary Fig. 7a). These embryos developed normally until the blastocyst stage (Supplementary Fig. 7b), with embryo-to-embryo and cell-to-cell variations in H3.3K36M expression (Supplementary Fig. 7a), likely due to incomplete Cre-mediated recombination. However, we recovered no *H3f3b*^K36M/+ embryo at E6.5, indicating that zygotic H3.3K36M leads to peri-implantation lethality (Supplementary Fig. 7c). This is reminiscent of the previously reported peri-implantation lethality of *Nsd1* KO mice[35]. In any case, as

maternal and zygotic H3.3K36M lead to similar phenotypes, it is currently unclear whether the loss of H3K36me2 in oocytes alone leads to the development phenotype.

## Loss of H3K36me3 leads to an excessive gain of CG methylation in H3K36me2-enriched regions

H3K36me3 deposition on chromatin is mediated by SETD2[36]. A previous study by *Setd2* knockout (KO) revealed a critical role of H3K36me3 in CG methylation in oocytes[14]. We performed WGBS on FGOs from our conditional *Setd2* KO (*Setd2*^flox/flox; *Zp3*-Cre) mice[37]

**Fig. 2 | Loss of H3K36me2 causes CG hypomethylation in MMRs of FGOs.**
**a** Representative images of control and *K36M* FGOs immunostained for H3K36me2 (left) and plots showing their signal intensities (right). Expression of HA-tagged H3.3K36M (H3.3K36M-HA) was confirmed by HA immunostaining. Signal intensity was measured in the control (*n* = 10) and *K36M* FGOs (*n* = 10) from two independent experiments. Scale bar, 10 μm. *P*-values (two-tailed Mann–Whitney U tests) are indicated. Error bars indicate the mean ± SD. Source data are provided as a Source Data file. **b** Heatmaps showing genome-wide H3K36me2 (left) and H3K36me3 enrichment (right) in 10 kb bins in control and *K36M* FGOs. Enrichment values were normalized using the spike-in control. **c** Genome browser snapshots showing H3K36me2 and H3K36me3 enrichment and CG methylation in control and *K36M* FGOs. A representative region on the X chromosome. CG methylation is lost upon H3K36me2 depletion in the regions highlighted in violet. The enrichment values were ChIP/input. **d** Scatter plots showing CG methylation levels in control and *K36M* FGOs. Fifty thousand randomly selected 10 kb bins were plotted with a color

gradient for H3K36me2 enrichment in control FGOs. **e** Violin plots showing CG methylation levels in control and *K36M* FGOs across the genome (left), autosomes (middle), and X chromosome (right). Horizontal bars indicate mean values. Pie charts show percentages of 10 kb bins categorized as HMRs, MMRs, and LMRs. **f** Boxplots showing CG methylation differences between *K36M* and control FGOs in individual chromosomes. CG methylation differences were determined in 10 kb bins (*n* = 19,191, 17,638, 15,612, 15,010, 14,673, 14,592, 13,734, 12,541, 12,115, 12,685, 11,859, 11,646, 11,673, 11,848, 10,067, 9495, 9155, 8737, and 5813 bins for chromosome 1, 2, …, and 19, respectively, and *n* = 15,494 bins for chromosome X). The box shows the median value and 25–75th percentiles, and whiskers show 1.5× interquartile range from the box. **g** Genome browser snapshots showing H3K36me2 enrichment in control FGOs and changes in CG methylation in *K36M* FGOs. CG methylation differences between *K36M* and control FGOs (ΔCGme) are indicated. The upper and lower panels indicate the representative regions of chromosomes 15 and X, respectively. H3K36me2 enrichment is indicated by ChIP/input.

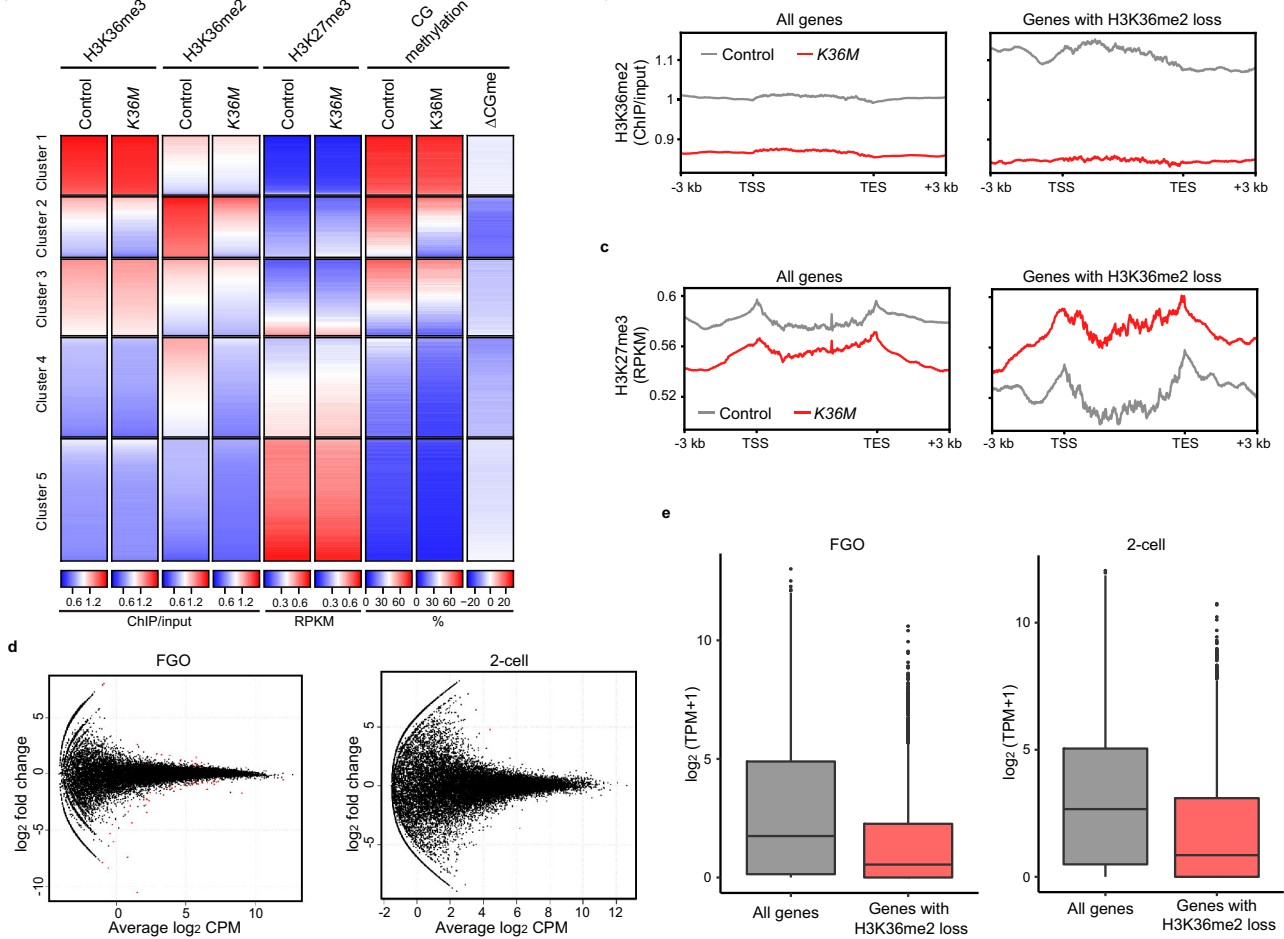

**Fig. 3 | Loss of H3K36me2 has a limited effect on the transcriptomes of FGOs and two-cell embryos.** **a** Heatmaps showing H3K36me3, H3K36me2, and H3K27me3 enrichment and CG methylation levels in control and *K36M* FGOs. CG methylation differences between *K36M* and control FGOs (ΔCGme). 10 kb bins from the whole genome were grouped into five clusters based on H3K36me2 and H3K36me3 enrichment statuses in control FGOs. **b** Plots showing H3K36me2 enrichment around genic regions in control and *K36M* FGOs. The analysis was performed for all genes (left, *n* = 23,081) and those with H3K36me2 loss (right, *n* = 2407). **c** Plots showing H3K27me3 enrichment around genic regions in control and *K36M* FGOs. The analysis was performed for all genes (left, *n* = 23,081) and

those with H3K36me2 loss (right, *n* = 2407). **d** MA plots showing changes in gene expression between *K36M* and control FGOs (left, *n* = 3) and between *K36M* oocyte-derived and control late two-cell embryos (right, *n* = 8). Differentially expressed genes with false discovery rate (FDR) < 0.05, are colored in red. CPM, counts per million. **e** Boxplots showing expression levels of all genes (*n* = 23,081) and genes with H3K36me2 loss (*n* = 2407) in control FGOs (left) and two-cell embryos (right). Genes with H3K36me2 loss included weakly expressed genes, such as olfactory receptor (*Olfr*) and vomeronasal receptor (*Vmnr*) genes (Supplementary Data 1). The box shows the median value and 25–75th percentiles, and whiskers show 1.5× interquartile range from the box.

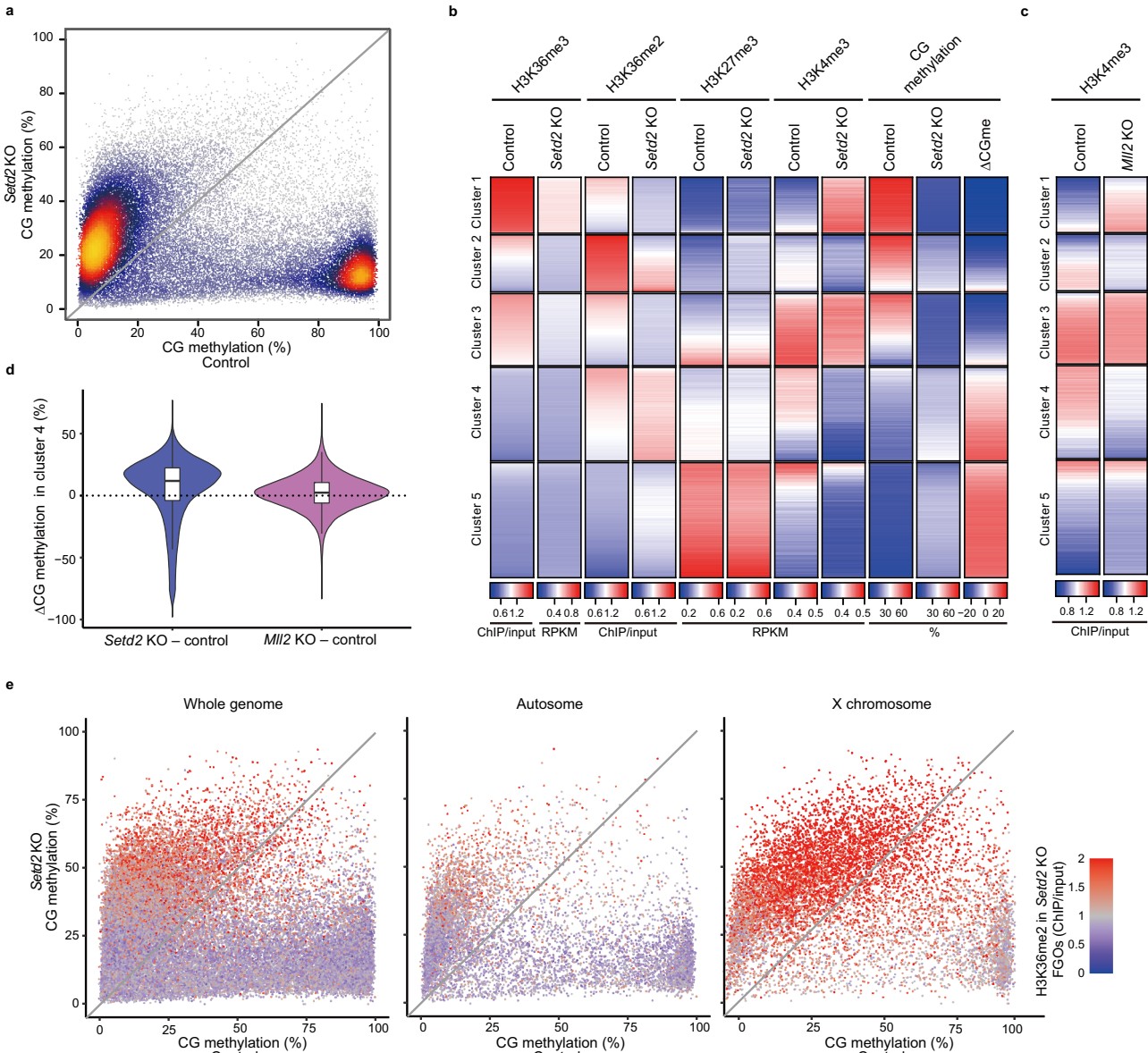

**Fig. 4 | Loss of H3K36me3 results in an excessive gain of CG methylation in genomic regions with specific histone marks. a** Scatter plots showing CG methylation levels in control and *Setd2* KO FGOs. Fifty thousand randomly selected 10 kb bins were plotted. **b** Heatmaps showing H3K36me3, H3K36me2, H3K27me3, and H3K4me3 enrichment and CG methylation levels in 10 kb bins in control and *Setd2* KO FGOs[11,14]. CG methylation differences between *Setd2* KO and control FGOs (ΔCGme). The data were sorted in the same order as in Fig. 3a. **c** Heatmaps showing H3K4me3 enrichment in 10 kb bins in control and *Mll2* KO FGOs[12]. **d** Violin plots

showing CG methylation differences in 10 kb bins of cluster 4 (*n* = 45,161) between *Setd2* KO and control FGOs (left) or between *Mll2* KO and control FGOs (right)[12]. Boxplots show median value and 25–75th percentiles, and whiskers show 1.5× interquartile range from the box. **e** Scatter plots showing CG methylation levels in control and *Setd2* KO FGOs across the genome (left), autosomes (middle), and X chromosome (right). Fifty thousand (left) or ten thousand randomly selected 10 kb bins (middle and right) were plotted with a color gradient for H3K36me2 enrichment in *Setd2* KO FGOs[11].

(the *Zp3* promoter drives Cre in an oocyte-specific manner[29]) (Supplementary Table 3) and confirmed a drastic decrease in CG methylation in HMRs (Fig. 4a). However, a gain of methylation occurred in regions lacking H3K36me3 in all chromosomes[14] (Fig. 4a and Supplementary Fig. 8a). The ChIP-seq data on *Setd2* KO and control FGOs published by others[11,14] (Supplementary Table 1) were reanalyzed to identify the histone mark associated with the gain of methylation. The methylation gain occurred in clusters 4 and 5 (Fig. 4b and Supplementary Fig. 8b; see Fig. 3a for clusters), many regions of which showed slightly increased H3K36me2 enrichment in *Setd2* KO FGOs[11]. The H3K27me3 enrichment was largely unchanged; however, the H3K4me3 level decreased in these clusters (Fig. 4b)[14]. H3K36me2 and

H3K4me3 have opposing roles in CG methylation;[5,15,38–40] therefore, the changes in these marks could have facilitated the methylation gain. To elucidate the role of H3K4me3, we examined CG methylation changes in *Mll2* KO oocytes using published WGBS data[12] (Supplementary Table 1), which have a partial depletion of H3K4me3. Loss of H3K4me3 was observed in cluster 4; however, CG methylation was largely unaffected (Fig. 4c, d)[12], suggesting that loss of H3K4me3 was not sufficient for the gain of CG methylation. The gain of methylation in *Setd2* KO FGOs was prominent in the X chromosome (Fig. 4e), which had strong H3K36me2 enrichment[11]. These observations strongly suggest the involvement of H3K36me2 in an excessive gain of CG methylation in *Setd2* KO FGOs.

## Simultaneous removal of H3K36me2 and H3K36me3 results in global hypomethylation in oocytes

To examine the impact of the combined loss of H3K36me2 and H3K36me3, we produced 'double mutant (*DM*)' female mice, in which H3.3K36M was expressed on a *Setd2* KO background (*Setd2*flox/fox; *H3f3b*K36M-flox/+; *Gdf9*-Cre). *DM* FGOs showed global loss of both H3K36me2 and H3K36me3 (Supplementary Fig. 9a, b). WGBS on *DM* FGOs revealed global hypomethylation (CG methylation level of 7.5%, compared to 36.4% in control FGOs) (Fig. 5a) (Supplementary Table 3), which was comparable to that of *Dnmt3a* KO FGOs (6.5%)[7] (Fig. 5b–d and Supplementary Fig. 9c; also see Fig.5a and Supplementary Fig. 9d) (Supplementary Table 1). Global hypomethylation was also confirmed using immunostaining for 5mC (Supplementary Fig. 9e). The precise methylation level and WGBS pattern were, however, slightly different between *DM* and *Dnmt3a* KO FGOs. As DNMT3A and DNMT3L proteins were normally expressed in *DM* FGOs (Fig. 5e), the observed hypomethylation was not secondary to the changes in the amount of these proteins. These results show that de novo CG methylation mediated by the DNMT3A-DNMT3L complex is heavily dependent on H3K36me2 and H3K36me3 in oocytes. Of note, the gain of CG methylation in *Setd2* KO FGOs was canceled in *DM* FGOs (see Figs. 4a and 5a), suggesting that H3K36me2 underlies this phenomenon. To further verify the importance of the DNMT3A-H3K36me2/3 axis in CG methylation of oocytes, we extracted a small number of genomic regions unaffected by *DM* and examined their CG methylation levels in *Dnmt3a* KO FGOs[7]. These regions were clearly unaffected in *Dnmt3a* KO FGOs (Fig. 5f). We also found that, conversely, regions unaffected in *Dnmt3a* KO FGOs were also unaffected by *DM* (Supplementary Fig. 9f). Lastly, we found that these regions were already CG methylated in nongrowing oocytes (Fig. 5g and Supplementary Fig. 9g), indicating that they are not subject to de novo CG methylation by DNMT3A and DNMT3L. In summary, H3K36me2 and H3K36me3 together provide an essential platform for guiding DNMT3A-dependent de novo DNA methylation in mouse oocytes (Fig. 5h).

## Discussion

Our study shows that H3K36me2 and H3K36me3 jointly provide the chromatin platform essential for guiding DNMT3A-dependent de novo DNA methylation in mouse oocytes. This is consistent with the fact that the PWWP domain of DNMT3A binds to both H3K36me2 and H3K36me3[10,15]. Furthermore, we found that the two histone marks are responsible for different levels of DNA methylation: H3K36me2, which is distributed broadly in intergenic regions and weakly transcribed regions, for moderate levels of DNA methylation, and H3K36me3, which essentially marks the actively transcribed regions, for high levels of methylation. How the two histone marks lead to different levels of DNA methylation is currently unknown, but the differential binding affinity of DNMT3A to the marks[10] and co-occurrence of other histone marks may play a role.

An interesting finding is a remarkable H3K36me2 enrichment in the X chromosome in oocytes: this is mostly attributed to a greater occupancy of H3K36me2-enriched domains, rather than a higher enrichment level in each domain, compared to the autosomes. We speculate that this H3K36me2 enrichment pattern may be related to the other findings of our study: unique localization of this chromosome in the nuclear periphery and tight association with the nuclear lamina. As lamina association can alter chromatin accessibility and chromatin-protein interaction[41], the unique spatial arrangement of the X chromosomes in oocytes could affect the creation of its distinct epigenetic signatures. Another important question is the biological significance of this unique epigenetic signature. In mice, the maternally derived X chromosome is imprinted so that it stays active in early embryos and in the extraembryonic lineages: this is achieved by posing a repressive histone mark H3K27me3 on the *Xist* locus[42], which is essential for the initiation of X chromosome inactivation, during

oocyte growth. Thus, the unique H3K36me2 enrichment could be somehow linked to this imprinted activation/inactivation. Alternatively, the broad H3K36me2 enrichment could be linked to the double dose of expression of the genes on the active maternal X chromosome, equalizing the dosage between the X chromosome and autosomes[43]. However, we did not see any sex-biased phenotype or preferential misregulation of the *Xist* or other X-linked genes. Therefore, the mechanism and biological significance of the preferential H3K36me2 enrichment in the X-chromosome warrant further investigation.

The ATRX-DNMT3A-DNMT3L (ADD) domain, which is commonly possessed by DNMT3A and DNMT3L and recognizes H3K4me0, could also contribute to chromatin recruitment of the protein complex[38,39] since DNA methylation is severely compromised in oocytes lacking KDM1B, an H3K4 demethylase[8,44]. Taking all these points into consideration, we speculate that H3K36me2 and H3K36me3 first provide the primary platform for recruitment and stable association of the DNMT3A-DNMT3L complex with its target regions. Then, upon recruitment, the ADD domains of DNMT3A and DNMT3L recognize H3K4me0, leading to the disruption of the autoinhibitory structure of DNMT3A[5,40]. While the DNMT3A PWWP domain does bind to H3K36me2 and H3K36me3, we recently found that DNA methylation still occurs in H3K36me2/3-enriched regions even when a point mutation (D329A) disrupting H3K36me2/3 binding is introduced[45]. Therefore, the DNMT3A-DNMT3L complex could have an additional, uncharacterized mechanism for recognizing H3K36me2/3. The epigenetic landscape of oocytes is crucial for embryonic development; therefore, further investigation of the precise mechanism of the proper DNMT3A-DNMT3L recruitment is warranted.

## Methods

### Mice

All animal experiments were approved by the Animal Experiments Committee of Kyushu University (A-20-104) and performed according to the guidelines for animal experiments at Kyushu University. Mice were housed in cages under specific pathogen-free conditions and had free access to water and food. *H3f3b*K36M-flox and *Setd2*flox mice[22,37] were maintained on C57BL/6 J;129 mixed and C57BL/6 J background, respectively.

### Oocyte collection, in vitro fertilization, and embryo culture

FGOs were collected by punching the ovaries of female mice at 3–4 weeks of age. Wild-type, control, *K36M*, *Setd2* KO, and *DM* FGOs were obtained from C57BL6/J, *H3f3b*K36M-flox/+, (*H3f3b*K36M-flox/+; *Gdf9*-Cre), (*Setd2*flox/flox; *Zp3*-Cre), and (*H3f3b*K36M-flox/+; *Setd2*flox/flox; *Gdf9*-Cre) females, respectively. Both *Gdf9* and *Zp3* promoters are oocyte-specific, but, for the *H3f3b*K36M-flox allele, *Gdf9*-Cre worked more efficiently, perhaps because it starts to be expressed earlier during oogenesis[29]. The primers used for genotyping are listed in Supplementary Table 4. For super-ovulation, females were injected with 0.1 ml of CARD HyperOva (Kyudo) and 46–48 h later, with 5 IU of hCG (ASKA Pharmaceutical). Cumulus-oocyte complexes were collected from the oviducts, and in vitro fertilization was performed with either C57BL/6 J or JF-1 sperm. Zygotes were cultured in EmbryoMax KSOM medium (Merck Millipore) at 37 °C under 6% $CO_2$ in the air. Late two-cell embryos were harvested at 31–32 h postfertilization (hpf) and blastocysts at 120 hpf.

### Fertility test

Females that were 8 weeks or older were placed together with a male in the same cage. Vaginal plugs were checked the following morning. The male was removed after a plug was observed, and the females were maintained for 20 days. Females that bore live pups were considered fertile.

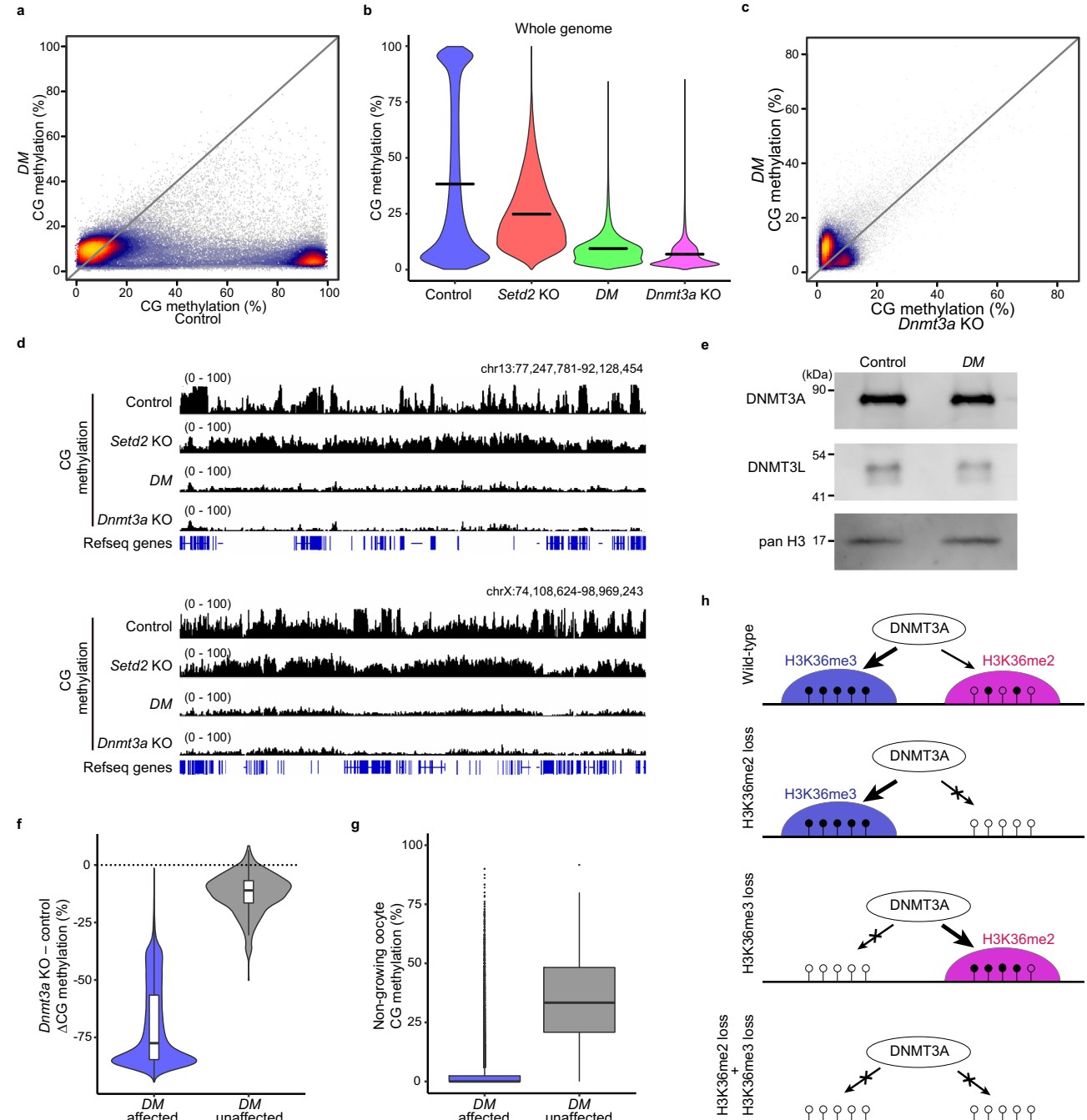

**Fig. 5 | Simultaneous removal of H3K36me2 and H3K36me3 results in global hypomethylation in FGOs. a** Scatter plots showing CG methylation levels in the control and *DM* FGOs. Fifty thousand randomly selected 10 kb bins were plotted. **b** Violin plots showing CG methylation levels in 10 kb bins across the genome in control, *Setd2* KO, *DM*, and *Dnmt3a* KO[7] FGOs. Horizontal bars indicate mean values. **c** Scatter plot showing CG methylation levels in *Dnmt3a* KO[7] and *DM* FGOs. Fifty thousand randomly selected 10 kb bins were plotted. **d** Genome browser snapshots showing CG methylation levels in control, *Setd2* KO, *DM*, and *Dnmt3a* KO[7] FGOs. The upper and lower panels show representative regions from chromosomes 13 and X, respectively. **e** Western blotting detecting DNMT3A and DNMT3L proteins in *DM* FGOs. Representative images from three biological replicates. Pan-H3 was used as the loading control. Source data are provided as a Source Data file. **f** Violin plots showing CG methylation differences between *Dnmt3a* KO[7]

and control FGOs in 10 kb bins of *DM* affected (*n* = 59,487) and *DM* unaffected regions (*n* = 450). Boxplots show median value and 25–75th percentiles, and whiskers show 1.5× interquartile range from the box. **g** Boxplots showing CG methylation levels in nongrowing oocytes[7] in 10 kb bins of *DM* affected (*n* = 59,487) and *DM* unaffected regions (*n* = 450). The box shows the median value and 25–75th percentiles, and whiskers show 1.5× interquartile range from the box. **h** A model summarizing H3K36me2/3-dependent CG methylation in mouse oocytes. H3K36me2 and H3K36me3 facilitate CG methylation in MMRs and HMRs, respectively (top). Upon H3K36me2 depletion, CG methylation is lost in MMRs (second), but upon H3K36me3 depletion, CG methylation is lost in H3K36me3-enriched regions and gained in H3K36me2-enriched regions (third). When both H3K36me2 and H3K36me3 are lost, CG methylation is severely lost (bottom). Open and filled circles indicate unmethylated and methylated CG sites, respectively.

## ULI-NChIP-seq

ULI-NChIP-seq was performed as described previously[46]. Three hundred cells were used per reaction, and 20% of each sample was taken as the input after MNase digestion. Samples were spiked with 1 µL of 0.216 pM control chromatin (SNAP-ChIP-Kmet Stat Panel, EpiCypher) to confirm the antibody specificity and normalize histone modification levels. The spike-in panel includes artificially synthesized histone H3 with various modifications, including mono-, di-, and tri-methylation at K36. For immunoprecipitation, 1 µg of H3K36me2 (Diagenode, cat. no. C15200182, 1:100) or H3K36me3 antibody (MBL, clone MABI0333, 1:100) were used. Ethanol-precipitated DNA was used for library preparation with NEBnext Ultra II DNA Library Prep Kit (NEB). All samples were subjected to PCR amplification for 15 cycles with KAPA HiFi Hot Start DNA polymerase (Kapa Biosystems). Paired-end sequencing was performed using an Illumina NovaSeq 6000 (53 bp ×2).

## CUT&RUN

CUT&RUN was performed with approximately 60 FGOs, as described previously[46]. In brief, samples were washed three times with CUT&RUN wash buffer, followed by incubation with an anti-H3K27me3 antibody (Diagenode, cat. no. C15410069, 1:100) in antibody incubation buffer overnight at 4 °C. After unbound antibodies were washed away, protein A-protein G-MNase was added and activated with $CaCl_2$. The libraries were subjected to paired-end sequencing on an Illumina NovaSeq 6000 (53 bp ×2).

## RNA-sequencing

FGOs and two-cell embryos were washed with PBS three times, flash frozen with liquid nitrogen, and stored at −80 °C until use. Each replicate contained five FGOs or a single late two-cell embryo. An SMART-Seq stranded kit (TAKARA) was used to prepare the RNA-seq libraries. The first and second amplification steps were carried out for 10 + 10 cycles for FGO and for 10 + 13 cycles for two-cell embryos. Paired-end sequencing was performed using an Illumina NovaSeq 6000 (53 bp ×2).

## WGBS

WGBS libraries were constructed using the post-bisulfite adaptor tagging method[47]. Pooled FGOs (150–200 cells per sample) were spiked with 1% unmethylated lambda phage DNA (Promega). Libraries were amplified using the KAPA library amplification kit (KK2620, KAPA) for four cycles. The resulting libraries were sequenced on an Illumina NovaSeq 6000 (108 bp, single-end).

## ChIP-seq and CUT&RUN data processing and analysis

ChIP-seq and CUT&RUN paired-end reads were aligned to the mouse genome (mm10) using Bowtie 2 (version 2.3.5.1)[48] with a default setting after removing adaptor sequences and low-quality reads and bases using Trim Galore! (version 0.6.2, Babraham Institute). Reads from PCR duplicates were removed using Samtools (version 1.9)[49] 'markdup' with the option '-r'. The numbers of ChIP and input reads covering each gene were counted using featureCounts (version 1.5.2)[50]. To compute the normalized enrichment for each bin, normalized ChIP read counts were divided by normalized input read counts using bamCompare in deepTools (version 3.4.3)[51] with options '--normalizeUsing RPKM' and '--operation ratio'. Data downloaded from the public database were normalized to their corresponding input data when available. The total number of reads mapped to the spike-in chromatin (SNAP-ChIP-Kmet Stat Panel, EpiCypher) was used to normalize the read counts in samples of different genotypes (Supplementary Table 2). (The number of reads from the spike-in was not sufficient for H3K36me2 ChIP-seq replicate 2; therefore, normalization was performed for replicate

1 only). Reads mapped to the spike-in chromatin containing histone H3 mono-, di-, or tri-methylated at K36 were used to check antibody specificity (Supplementary Table 2). Heatmaps were depicted using computeMatrix and plotHeatmap in deepTools (version 3.4.3)[51]. For heatmap analysis, we classified 10 kb bins into five clusters based on H3K36me2 and H3K36me3 enrichment status in control FGOs. The following genes were considered to have lost H3K36me2 in the gene body regions: H3K36me2 ChIP/input > 1 in control FGOs and ChIP/input ratio between K36M versus control FGOs < 0.67 (over 33% reduction).

## Lamina-associated domain data analysis

Reads obtained from Dam-Lamin B1 and Dam expression were aligned to the mouse genome (mm10) using Bowtie 2 after removing adaptor sequences and low-quality reads and bases using Trim Galore!. Duplicate and low-mapping-quality (MAPQ < 5) reads were removed to calculate reads per kilobase per million (RPKM) values in 10 kb bins. The Dam-Lamin B1 data were normalized to their corresponding Dam-only data.

## RNA-seq data analysis

For samples prepared using a SMART-Seq Stranded Kit (Takara), the first three bases of Read 2 derived from the SMART-seq Stranded Adaptor were removed using Trim Galore! before mapping with Hisat2 (version 2.2.0)[52]. Reads corresponding to ribosomal RNA were removed using a BEDTools intersectBed function[53]. FeatureCounts was used to analyze gene expression. The resulting read count data were processed using EdgeR[54] to identify differentially expressed transcripts between datasets. A false discovery rate of <0.05, was used to extract differentially expressed transcripts. Replicate 4 of the control two-cell embryo indicated a technical problem; therefore, it was not used for analysis. To distinguish transcripts derived from the maternal and paternal alleles in two-cell embryos, strain-specific single nucleotide polymorphisms (SNPs) reported for C57BL/6 J (mm9) and JF-1[55] mice were converted into mm10-based data using LiftOver (UCSC Genome Browser) (version 432). We maintained $H3f3b^{K36M-flox}$ mice under C57BL/6 J;129 background, SNPs common to JF-1 and 129 were excluded. To obtain a modified mm10 fasta file (SNPs are N-masked) for allelic analysis, we utilized BEDTools (version 2.29.0) 'maskfasta'. Trimmed fastq data were then aligned to the reference in which the SNPs were N-masked using Hisat2. Finally, the output BAM files were processed with SNPsplit (version 0.3.2) to classify every SNP-containing read as either maternal or paternal.

## WGBS data analysis

Reads were trimmed to remove adaptor sequences and low-quality reads and bases using Trim Galore! and mapped to the mouse genome (mm10) using Bismark (version 0.14.5)[56]. After confirming reproducibility between replicates, BAM files were merged using Samtools 'merge'. CG sites covered by four to 100 reads were considered informative, and bins containing more than five informative CG sites were used for downstream analyses using custom Perl scripts (version 5.16.3). To extract bins affected and unaffected by DM or Dnmt3a KO in FGOs, we used the following criteria: affected, ΔCG methylation (mutant − control) < −20% and the CG methylation level in control > 40%; unaffected, −10% ≤ ΔCG methylation (mutant − control) < 10% and the CG methylation level in control ≥ 40%.

## Immunofluorescence

Immunofluorescence staining of FGOs and blastocysts was performed as described previously[46], except that the specimens were fixed with 4% paraformaldehyde prior to permeabilization with 0.2% Triton X-100. For 5mC detection, fixed and permeabilized zona-free oocytes were treated with 4 N HCl for 30 min at room temperature. They were

then treated with 100 mM Tris·HCl (pH 8) for 10 min at room temperature and incubated in a blocking buffer. FGOs or blastocysts were incubated in blocking buffer containing primary antibodies overnight at 4 °C. Primary antibodies used were anti-H3K36me2 (Cell Signaling Technology, cat. no. 2901, 1:500), anti-H3K36me2 (Diagenode, cat. no. C15200182, 1:500), anti-H3K36me3 (Cell Signaling Technology, cat. no. 4909, 1:500), anti-H3K36me3 (Active Motif, cat. no. 61101, 1:1000), anti-HA (abcam, cat. no. ab9110, 1:500), anti-HA (Biolegend, cat. no. 901513, 1:500), and anti-5mC (Merck Millipore, cat. no. NA81-50UGCN; 1:1000 dilution) antibodies. The secondary antibodies were Alexa488-conjugated donkey anti-mouse IgG, Alexa555-conjugated donkey anti-rabbit IgG, Alexa488-conjugated goat anti-mouse IgG, and Alexa594-conjugated goat anti-rabbit IgG antibodies (Thermo Fisher Scientific, cat. no. A-21202, A-31572, A-11001, and A-11037, 1:500). For acquisition of images by LSM-700 or LSM-900 confocal microscope (ZEISS) using Zen software (black edition for LSM-700, version 14.0.24.201; blue edition for LSM-900, version 3.1), parameters were first set to avoid signal saturation, and the same setting was used for all specimens. When the genotype could be identified using immunofluorescence for HA, specimens of different genotypes were processed together to minimize technical variations[46].

### Fluorescence intensity measurement
Nuclear areas were first determined by thresholding of DAPI staining images using Fiji software (version 1.52e)[57]. The intensity of the DAPI signal was normalized. Data obtained from independent experiments were combined after examining their reproducibility. A two-tailed Mann–Whitney U test was used to compare the experimental groups. Maximum intensity projection images were also obtained using the Fiji software.

### SDS-PAGE and western blotting
Fifteen FGOs from mice at 3–4 weeks of age were harvested and boiled at 95 °C for 5 min in 10 µL of Laemmli sample buffer. SDS-PAGE was conducted on a 5–20% gradient gel (Nacalai Tesque). For western blotting, proteins were transferred to a polyvinylidene difluoride membrane using a wet tank system. For immunoblotting of proteins with large molecular weights, proteins were transferred at 20 V for 8 h at 0 °C. After blocking with 5% skim milk in Tris-buffered saline with 0.01% Tween20 (TBS-T), immunoblotting was performed with primary and secondary antibodies diluted in Can Get Signal solutions 1 and 2 (TOYOBO), respectively. The antibodies used were rabbit anti-DNMT3A (Abcam, cat. no. ab188470, 1:1000), rabbit anti-DNMT3L (abcam, cat. no. ab194094, 1:1000), rabbit anti-pan-H3 (Abcam, cat. no. ab1791, 1:1000), and goat anti-rabbit IgG H&L (HRP) antibodies (Abcam, cat. no. ab6721, 1:10000).

### Quantification and statistical analysis
We confirmed the reproducibility of the replicate data (Supplementary Note), and representative results are shown where appropriate. For H3K36me3 ChIP-seq data and WGBS data from *Setd2* KO FGOs, we analyzed only one pool of specimens, but compared the data with published data to validate the results[14] (Supplementary Note). Graphic representation of data and statistical tests were performed using R software (version 3.6.1) (http://www.r-project.org) or Excel 2013. Statistical significance was set at $p < 0.05$.

## Data availability
The data that support this study are available from the corresponding authors upon reasonable request. The sequencing data from this study are available at the Gene Expression Omnibus under accession code GSE183969. Publicly available data reanalyzed in this study are listed in Supplementary Table 1. They are available from GSE112551, GSE148150, GSE112835, GSE93941, and DRA000570. The mouse reference genome data (mm10) and the SNP information for the JF-1 mouse genome are available from https://hgdownload.cse.ucsc.edu/goldenpath/mm10/ and https://molossinus.brc.riken.jp/pub/For_Seq_Analysis/list_of_variations/, respectively. Source data are provided with this paper.

## Code availability
Custom code used to process and analyze the genomic data, as detailed in the Methods, are available from the corresponding authors upon reasonable request.

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

## Acknowledgements

We thank K. Hayashi (Kyushu University) for sharing a confocal microscope and K. Shirane (Osaka University) for comments on the manuscript. We also thank the members of our laboratory, especially W.K. Au Yeung, J. Oishi, T. Hanagiri, and M. Miyake, and the common research facilities of the Medical Institute of Bioregulation, Kyushu University, for technical assistance. This work was supported by grants from a MEXT Grant-in-Aid for Scientific Research on Innovative Areas (JP19H05756 to T.I.); NIH grants (R35GM141085 to S.H.N.), (R01CA248019 and R01DK130478 to G.H.); and a JSPS Grant-in-Aid for Specially Promoted Research (JP18H05214 to H.S.).

## Author contributions

T.I. and H.S. conceived the project, and T.I., Y.O., and H.S. jointly supervised the project. S.Y. and T.I. designed the experiments, performed immunofluorescence, and prepared samples for sequencing. S.Y. analyzed all the data. S.A., S.N., and G.H. contributed to establish the mouse lines used in this study. S.Y., T.I., and H.S. interpreted the data and wrote the manuscript with contributions from all authors.

## Competing interests

The authors declare no competing interests.
