## [Peer Review File · Nature Communications]

REVIEWER COMMENTS

Reviewer #1 (Remarks to the Author):

The PWWP domain of the mammalian de novo DNA methyltransferases, DNMT3A and 3B, is a reader of H3K36 methylation, which helps direct DNA Methylation establishment. In the past few years, studies in mice have shown that the DNMT3A PWWP has a stronger affinity for H3K36me2 (found mainly in intergenic regions), and that of 3B for H3K36me3 (gene bodies). Consistent with this, in sperm, where DNMT3A is the primary de novo methyltransferase, loss of H3K36me2 leads to a dramatic reduction in DNA Methylation (PMID 32929285). Curiously, in oocytes DNMT3A is also the major DNMT, however DNA Methylation is mainly found in gene bodies and loss of H3K36me3 leads to a huge DNA Methylation defect (PMID 31040401). Here, Yano et al revisit the role of H3K36me2 in fully grown oocytes (FGOs) in mice by using an oncohistone (H3K36M dominant negative mutation) that reduces H3K36me2 but not -me3. There are moderate DNA Methylation defects that are observed, indicating the mark does play a role. Curiously, there is a more pronounced effect on the X chromosome than on autosomes, although this observation is never really explained. Strikingly, although the K4M mutant seems to have a minimal effect on oocyte transcription, it renders the oocytes inviable. Once again this is not fully explained (see major comment). Finally, a Setd2 mutation (K36me3 MTase) combined with the K36M mutation leads to a near-complete loss of DNA Methylation, mimicking the Dnmt3A mutant phenotype. This is the strongest experiment to demonstrate the role of H3K36me2, as indeed there is a substantial amount of residual DNA Methylation left in Setd2 mutants. It indicates that K36me2 does play a role in targeting DNA Methylation, and the genetic cross implies that proper deposition of the mark is important.

Overall, I really appreciated this paper. The observations will certainly be of interest to the mammalian epigenetics field, as the relationship between H3K36 methylation and DNA Methylation still being understood at the mechanistic and developmental level. The quality of the data is high, and the results are clearly and succinctly explained. The study also raises interesting questions for future research. I have two major comments and some minor comments that I would like to see addressed for a revised version.

Major comment

- I found the section on embryonic lethality (supp figure 6) to be tantalizing but incomplete. The authors claim that the H3K36me2 deficit in the oocyte leads to the lethality, but this is not demonstrated nor explained. The in vitro fertilization experiment was not sufficient. One experiment that could be performed to demonstrate the role of the maternal contribution is a proper cross where the K36M mutation is expressed only in the early embryo. Alternatively, injecting the H3K36me2 demethylase (eg, KDM8) into zygotes might be a more straightforward way to achieve this same end—a similar strategy has been used with success by Yi Zhang's lab. If the embryos survive past implantation, that would bolster the claims in this section. It might be difficult to show **why** maternal K36me2 plays such an important role, given the minimal effect on the oocyte transcriptome, but it should at least be shown definitively it is the case.
- Previous studies have made the link between transcription and DNA Methylation in the oocyte (eg, PMID 26408185) . This also ties in with H3K36me3, as this mark is deposited concomitantly with transcriptional elongation. The authors commented on this in the introduction. I think adding previously transcriptome data (either published or from this study) would therefore be useful. Does H3K36me2-dependent methylation fall outside of transcriptionally active regions? Is it that clear? This deserves some proper analysis and a figure.

Minor comments

- I would appreciate more discussion about the bizarre patterns exclusively observed on the X chromosome. Why is this chromosome such an outlier, given what's known about regulation in preimplantation development. And why would DNA Methylation and H3K36me2 exhibit different patterns there? This should be commented on in the discussion section.
- Could the authors add a supplemental figure containing a gene model of the K36M construct used for this work? I had to look in other studies to find it, and this would be useful for the reader.
- It looks like from the screenshots there is some residual H3K36me2 in gene bodies in the K36M mutant (eg, Fig 2c). Is this genuine K36me2 (and if so why?) or perhaps antibody cross-reactivity with H3K36me3?

Reviewer #2 (Remarks to the Author):

The authors use a conditional H3K36M mutant expressed in mouse oocytes to study the downstream effects of depletion of the histone H3K36me2 modification. They first show that intergenic H3K36me2 is associated with moderate levels of CpG methylation – significantly lower than genic DNAm associated with K36me3. Depletion of H3K36me2 results in moderate reduction of DNAm, which is largely limited to regions depleted of H3K36me2. In contrast, depletion of H3K36me3 results in loss of DNAm within expressed genic regions, and a gain of DNAm in intergenic compartments. Expression of H3K36M histone along with a KO of SETD2 results in the greatest reduction of DNAm, which the authors note to be comparable with a KO of DNMT3A. Embryos resulting from the H3K36M expressing oocytes exhibit pre-implantation lethality. Finally, the authors note differential effects of depletion H3K36me2 on autosomes vs the X-chromosomes.

I find that the analysis is well performed, and I agree with most of the conclusions drawn by the authors. The authors produce new data, but they also use primary data from previous studies to augment their analyses. Overall, this is a valuable contribution to our knowledge of the role of histone modifications, particularly H3K36me2/3, on establishing methylation patterns in oocytes.

On the critical side, I find the novelty of the manuscript to be limited. This study follows two recent papers, Xu et al, Nature Genetics 2019 (showing the requirement of SETD2-deposited H3K36me3 in mouse oocytes through guiding DNA methylation patterns), and Shirane et al, Nature Genetics 2020 (showing a similar requirement for NSD1 deposited H3K36me2 in the male germline). Here, the authors extend the previous findings showing that H3K36me2 is relevant not only in male, but also female germline. Notably, the effect of H3K36me2 in oocytes is lower than in sperm, presumably because of the relatively lower levels of this mark.

This observation is original – in the context of oocytes – but again, it follows a number of previous studies that demonstrated the recruitment of DNMT3A (and to some extent DNMT3B) by NSD1 or NSD2 deposited H3K36me2 in a number of contexts. These include: prospermatogonia (Shirane et al. Nat Gen 2020), mESCs and mMSCs (Weinberg et al. Nat Gen 2019), in vitro and mouse cell lines (Dukatz et al. JMB 2019), squamous carcinoma (Farhangdoost et al. Cell Rep 2021), cancer cell lines (Xu et al., Protein and Cell 2020). As a consequence, the finding of the effect of H3K36me2 on DNAm is new in the specific context of mouse oocytes, but has been observed more generally before.

The observation which I found to be novel and intriguing, is the differential effect of

H3K36me2 on the X chromosomes and autosomes. I was disappointed to see that the authors did not follow this further.

Specific Points:

1) In several instances, I found the methods descriptions to be too terse and requiring a lot of effort on my part to understand what is happening. E.g. the description of the mouse models is minimal – while this description is provided in the previous work by some of the authors, it was not immediately clear to me which were the appropriate references to follow. Please use a clearer approach, such as “as described previously in (ref)”.

2) Again, on the methods front, a clearer description of the system used, including a graphic would be immensely helpful. I am not a germ cell biologist, and it took me unnecessarily long time to figure out at which stage of oocyte development zp3 or Gdf9 are expressed. Similarly, it took a eureka moment to realize that while H3K36M is expressed in all oocytes, only half of them will inherit the continuous expression of the mutation while the other half will only show effects of early effect of the mutant histone. A simple graphic would be great. The one included in the Supplementary Figure is not sufficient.

3) Please indicate clearly which data is produced here and which comes from previous studies. I was confused as to why so few datasets have been deposited in GEO.

4) Please provide the data for the SNAP-ChIP-Kmet Stat Panel. The manuscript mentions that it has been used for normalization, but the normalization factors themselves are not provided. It would be useful to have the raw data (number of reads for each spike in), since in our hands this spike-in approach has not worked according to expectations, and it would be great to share these types of numbers with the community.

The following points 5 and 6 are more major and less easily addressable:

5) The authors state and show evidence that the H3K36M mutation does not affect H3K36me3 levels. This is in contrast to previous studies that indicate that K36M affects all K36 KMTs, including SETD2 and H3K36me3 levels (e.g. Yang et al. Genes Dev 2016, Liu et al. Cell Discovery 2021, Zhang et al. Sci Rep 2017, Papillon-Cavanaugh et al. Nat Gen 2017, Brumbaugh et al. Nat Cell Biol 2019, and several others). Those papers concern cell lines, cancers, and mouse tissues. The authors observation that K36M mutant h3f3b does not affect k36me3 in their system needs to be at least discussed in view of the other results. Unfortunately, the previous findings shed some uncertainty in my mind as to whether the authors are actually only affecting H3K36me2 here, or whether some of their results – see also below – may be caused by other mechanisms.

6) The authors claim “Loss of H3K36me2 in oocytes causes embryonic lethality around implantation”, is weakened by the fact that they are expressing a mutant histone, with possibly other effects than inhibiting NSD enzymes. My first concern was that the tagged version of the histone may be detrimental. It took me some unnecessarily long searching to find out that the authors have already demonstrated that expression of the WT version of the tagged histone has no discernible phenotype – again prior work needs to be better indicated and explicitly referenced. Once I understood what is happening, I was impressed by the idea that half of the eggs should not continue to express K36M, and their future lethality is only affected by the early depletion of H3K36me2. However, that is not quite true, since the mutant histones will persist in the zygote, and even at the ~100 cell stage should be present at ~ 1/50 of the initial levels (even if not detectable by staining).

7) I was not convinced that the DM condition recapitulates DNMT3A KO methylation

patterns, as the authors imply. It does result in low levels of methylation, which is interesting, but the correlation plot in figure 5c clearly shows two clusters, one that is more affected by DNMT3A than DM and one that is more affected by DM than DNMT3A. This should be explained. Similarly, I don't understand the premise behind figure 5f and the rationale of identifying regions that don't change in one condition and showing that they change little in the other condition. Biologically, what does this mean and why should this happen? Quite often, such extreme set of regions represent some artifact of the analysis, rather than having any biological relevance.

8) The discussion is very terse and should be better used to position this paper within the existing literature, as well as to clarify some of the concerns mentioned above.

Addressing the above points will improve the manuscript, in my opinion. Overall, I think that the analysis is sound, and I appreciate the value of the work. However, the remaining concern will still be the moderate level of novelty for this level of publication.

Point-by-point Responses to the Reviewers:

Manuscript Number: NCOMMS-21-45097-T

Title: Histone H3K36me2 and H3K36me3 form a chromatin platform essential for DNMT3A-dependent DNA methylation in mouse oocytes

We appreciate the time and efforts that the reviewers put into reviewing our manuscript and find all comments helpful to improve our manuscript. According to their suggestions, we have performed additional experiments and analyses, the outcomes of which further support our conclusions. The only exception is the cause of the developmental phenotype: the additional experiment did not help resolve the question and it is left open for future studies. By incorporating the helpful comments from the reviewers, we have also revised the main text and figures, which we believe has improved the clarity and readability of our manuscript.

The following figure panels and tables have been revised or replaced by a new one.

Fig. 5f

Supplementary Fig. 6 (The figure title has been changed and the panels reordered.)

Supplementary Table 2 (Previous Supplementary Table 1)

The following figures, panels, and tables have been newly created and added.

Fig. 5g

Supplementary Fig. 7

Supplementary Figs 3a,b, 5c, and 9f,g.

Supplementary Tables 1, 5, and 6

Lastly, we have corrected some typos and grammatical errors. All these changes are shown in red in the revised manuscript. We hope that the manuscript is now suitable for publication in *Nature Communications*.

Below you will find our point-by-point responses to the reviewers.

[Reviewer #1]

The PWWP domain of the mammalian de novo DNA methyltransferases, DNMT3A and 3B, is a reader of H3K36 methylation, which helps direct DNA Methylation establishment. In the past few years, studies in mice have shown that the DNMT3A PWWP has a stronger affinity for H3K36me2 (found mainly in intergenic regions), and that of 3B for H3K36me3 (gene bodies). Consistent with this, in sperm, where DNMT3A is the primary de novo methyltransferase, loss of H3K36me2 leads to a dramatic reduction in DNA Methylation (PMID 32929285). Curiously, in oocytes DNMT3A is also the major DNMT, however DNA Methylation is mainly found in gene bodies and loss of H3K36me3 leads

to a huge DNA Methylation defect (PMID 31040401). Here, Yano et al revisit the role of H3K36me2 in fully grown oocytes (FGOs) in mice by using an oncohistone (H3K36M dominant negative mutation) that reduces H3K36me2 but not -me3. There are moderate DNA Methylation defects that are observed, indicating the mark does play a role.

Curiously, there is a more pronounced effect on the X chromosome than on autosomes, although this observation is never really explained. Strikingly, although the K4M mutant seems to have a minimal effect on oocyte transcription, it renders the oocytes inviable. Once again this is not fully explained (see major comment). Finally, a Setd2 mutation (K36me3 MTase) combined with the K36M mutation leads to a near-complete loss of DNA Methylation, mimicking the Dnmt3A mutant phenotype. This is the strongest experiment to demonstrate the role of H3K36me2, as indeed there is a substantial amount of residual DNA Methylation left in Setd2 mutants. It indicates that K36me2 does play a role in targeting DNA Methylation, and the genetic cross implies that proper deposition of the mark is important.

Overall, I really appreciated this paper. The observations will certainly be of interest to the mammalian epigenetics field, as the relationship between H3K36 methylation and DNA Methylation still being understood at the mechanistic and developmental level. The quality of the data is high, and the results are clearly and succinctly explained. The study also raises interesting questions for future research. I have two major comments and some minor comments that I would like to see addressed for a revised version.

Our response: We are pleased to know that this reviewer is positive.

Major comment

1) I found the section on embryonic lethality (supp figure 6) to be tantalizing but incomplete. The authors claim that the H3K36me2 deficit in the oocyte leads to the lethality, but this is not demonstrated nor explained. The *in vitro* fertilization experiment was not sufficient. One experiment that could be performed to demonstrate the role of the maternal contribution is a proper cross where the K36M mutation is expressed only in the early embryo. Alternatively, injecting the H3K36me2 demethylase (eg, KDM8) into zygotes might be a more straightforward way to achieve this same end—a similar strategy has been used with success by Yi Zhang's lab. If the embryos survive past implantation, that would bolster the claims in this section. It might be difficult to show *why* maternal K36me2 plays such an important role, given the minimal effect on the oocyte transcriptome, but it should at least be shown definitively it is the case.

Our response: We appreciate the reviewer's comments and constructive suggestions on the developmental phenotype. We have performed an additional experiment: H3.3K36M was expressed only in early embryos, not in oocytes, by crossing *Gdf9-Cre* females with *H3f3b*^{K36M-flox/flox} males. (While we agree that injecting a histone demethylase may be an alternative, the technology is not in our hands. Also, experts say that setting the appropriate condition depends on trial and error and thus requires time.) We confirmed the H3.3K36M expression in derived embryos and found that they

develop normally until the blastocyst stage (new **Supplementary Fig. 7a,b**). At E6.5, however, we recovered no embryo with the *H3f3b*^{K36M/+} genotype (new **Supplementary Fig. 7c**), indicating that zygotic H3.3K36M expression has a developmental outcome similar to maternal H3.3K36M expression. Thus, we were not able to resolve the maternal versus zygotic problem, and whether the loss of H3K36me2 in oocytes alone is detrimental to development remains as an open question. This is unfortunate, but the data does not exclude the role of maternal H3K36me2 in development. Anyway, as the major focus of this study is the mechanistic side, we should like to leave the question for future studies. We describe these findings in the results section and tone down our conclusion in our revised manuscript (**page 9, paragraph 2, L161-173**).

2) Previous studies have made the link between transcription and DNA Methylation in the oocyte (eg, PMID 26408185). This also ties in with H3K36me3, as this mark is deposited concomitantly with transcriptional elongation. The authors commented on this in the introduction. I think adding previously transcriptome data (either published or from this study) would therefore be useful. Does H3K36me2-dependent methylation fall outside of transcriptionally active regions? Is it that clear? This deserves some proper analysis and a figure.

Our response: In response to the reviewer's comment, we have performed an additional analysis of our transcriptome data. We found that, while H3K36me3-enriched domains (genes) are transcribed at high levels, H3K36me2-enriched domains (genes) show much lower levels of transcription, just as expected (new **Supplementary Fig. 5c**). We briefly touch upon this result in the results section (**page 8, paragraph 2, L134-137**).

Minor comments

3) I would appreciate more discussion about the bizarre patterns exclusively observed on the X chromosome. Why is this chromosome such an outlier, given what's known about regulation in preimplantation development. And why would DNA Methylation and H3K36me2 exhibit different patterns there? This should be commented on in the discussion section.

Our response: We thank the reviewer for this great suggestion. We have added a paragraph to give our speculation on why the X chromosome exhibits H3K36me2 and DNA methylation patterns that are different from the autosomes (**page 13, paragraph 2, L233-250**). We believe that this addition has improved the discussion section.

4) Could the authors add a supplemental figure containing a gene model of the K36M construct used for this work? I had to look in other studies to find it, and this would be useful for the reader.

Our response: We apologize that the experimental scheme was not clearly given and thank the reviewer for his/her helpful comment. We have added panels showing the K36M gene model and the timing of *Gdf9*-Cre expression (new **Supplementary Fig. 3a,b**). We hope that they will help the readers comprehend the experiment.

5) It looks like from the screenshots there is some residual H3K36me2 in gene bodies in the K36M mutant (eg, Fig 2c). Is this genuine K36me2 (and if so why?) or perhaps antibody cross-reactivity with H3K36me3?

Our response: The residual H3K36me2 signals can be explained by either incomplete inhibition of the H3K36 methyltransferases by H3.3K36M or cross-reactivity of the antibody. We have confirmed the high specificity of the H3K36me2 antibody using SNAP-ChIP-Kmet Stat Panel, the results of which are now included in **Supplementary Table 2** (Previous Supplementary Table 1). Thus, the cross-reactivity is an unlikely cause, but it is hard to exclude it completely. Anyway, the observed H3K36me2 loss was great and sufficient to cause a great loss of DNA methylation.

[Reviewer #2]

The authors use a conditional H3K36M mutant expressed in mouse oocytes to study the downstream effects of depletion of the histone H3K36me2 modification. They first show that intergenic H3K36me2 is associated with moderate levels of CpG methylation – significantly lower than genic DNAm associated with K36me3. Depletion of H3K36me2 results in moderate reduction of DNAm, which is largely limited to regions depleted of H3K36me2. In contrast, depletion of H3K36me3 results in loss of DNAm within expressed genic regions, and a gain of DNAm in intergenic compartments. Expression of H3K36M histone along with a KO of SETD2 results in the greatest reduction of DNAm, which the authors note to be comparable with a KO of DNMT3A. Embryos resulting from the H3K36M expressing oocytes exhibit pre-implantation lethality. Finally, the authors note differential effects of depletion H3K36me2 on autosomes vs the X-chromosomes.

I find that the analysis is well performed, and I agree with most of the conclusions drawn by the authors. The authors produce new data, but they also use primary data from previous studies to augment their analyses. Overall, this is a valuable contribution to our knowledge of the role of histone modifications, particularly H3K36me2/3, on establishing methylation patterns in oocytes.

On the critical side, I find the novelty of the manuscript to be limited. This study follows two recent papers, Xu et al, Nature Genetics 2019 (showing the requirement of SETD2-deposited H3K36me3 in mouse oocytes through guiding DNA methylation patterns), and Shirane et al, Nature Genetics 2020 (showing a similar requirement for NSD1 deposited H3K36me2 in the male germline). Here, the authors extend the previous findings showing that H3K36me2 is relevant not only in male, but also female germline. Notably, the effect of H3K36me2 in oocytes is lower than in sperm, presumably because of the relatively lower levels of this mark.

This observation is original – in the context of oocytes – but again, it follows a number of previous studies that demonstrated the recruitment of DNMT3A (and to some extent DNMT3B) by NSD1 or NSD2 deposited H3K36me2 in a number of contexts. These include: prospermatogonia (Shirane et al. Nat Gen 2020), mESCs and mMSCs (Weinberg et al. Nat Gen 2019), in vitro and mouse cell lines (Dukatz et al. JMB 2019), squamous carcinoma (Farhangdoost et al. Cell Rep 2021), cancer cell lines

(Xu et al., Protein and Cell 2020). As a consequence, the finding of the effect of H3K36me2 on DNAm is new in the specific context of mouse oocytes, but has been observed more generally before.

The observation which I found to be novel and intriguing, is the differential effect of H3K36me2 on the X chromosomes and autosomes. I was disappointed to see that the authors did not follow this further.

Our response: We appreciate the reviewer's critical comments. Keeping all these comments in mind, we have revised the manuscript as detailed below.

Specific Points:

1) In several instances, I found the methods descriptions to be too terse and requiring a lot of effort on my part to understand what is happening. E.g. the description of the mouse models is minimal – while this description is provided in the previous work by some of the authors, it was not immediately clear to me which were the appropriate references to follow. Please use a clearer approach, such as “as described previously in (ref)”.

Our response: We apologize that our method description was not comprehensible enough to readers outside the field. According to his/her suggestion, we now provide a brief description of our mouse models with an appropriate reference in the revised manuscript (**page 6, paragraph 2, L95-106**). Furthermore, we provide a panel that explains the models used in this study (new **Supplementary Fig. 3a,b**). (Also see our response to *Minor comment 4* of reviewer #1.)

2) Again, on the methods front, a clearer description of the system used, including a graphic would be immensely helpful. I am not a germ cell biologist, and it took me unnecessarily long time to figure out at which stage of oocyte development zp3 or GDf9 are expressed. Similarly, it took a eureka moment to realize that while H3K36M is expressed in all oocytes, only half of them will inherit the continuous expression of the mutation while the other half will only show effects of early effect of the mutant histone. A simple graphic would be great. The one included in the Supplementary Figure is not sufficient.

Our response: As mentioned above, we have added explanations of the mouse models in the results section (**Page 6, paragraph 2, L95-106**) (**Page 9, paragraph 1, L149-151; L154-160**) and methods section with appropriate references (**page 16, paragraph 2, L277-279**). We have also added and revised panels showing the experimental outline (new **Supplementary Fig. 3a,b, Supplementary Fig. 6a**, and new **Supplementary Fig. 7a**).

3) Please indicate clearly which data is produced here and which comes from previous studies. I was confused as to why so few datasets have been deposited in GEO.

Our response: All datasets shown in **Supplementary Tables 2 and 3** were produced in this study and deposited altogether in the GEO database under one accession number, as already described in our original manuscript. For published datasets, we clearly cited appropriate references in both main text and figure legends in the original manuscript. For readers' convenience, however, we newly

provide **Supplementary Table 1**, which summarizes published datasets used in this study.

4) Please provide the data for the SNAP-ChIP-Kmet Stat Panel. The manuscript mentions that it has been used for normalization, but the normalization factors themselves are not provided. It would be useful to have the raw data (number of reads for each spike in), since in our hands this spike-in approach has not worked according to expectations, and it would be great to share these types of numbers with the community.

Our response: We have described the normalization method in the methods section (**page 17, paragraph 3, L295-297; page 19, paragraph 1, L330-335**) and added the data in **Supplementary Table 2**. We hope that the data will be useful to the reviewer and other scientists.

The following points 5 and 6 are more major and less easily addressable:

5) The authors state and show evidence that the H3K36M mutation does not affect H3K36me3 levels. This is in contrast to previous studies that indicate that K36M affects all K36 KMTs, including SETD2 and H3K36me3 levels (e.g. Yang et al. Genes Dev 2016, Liu et al. Cell Discovery 2021, Zhang et al. Sci Rep 2017, Papillon-Cavanaugh et al. Nat Gen 2017, Brumbaugh et al. Nat Cell Biol 2019, and several others). Those papers concern cell lines, cancers, and mouse tissues. The authors observation that K36M mutant h3f3b does not affect k36me3 in their system needs to be at least discussed in view of the other results. Unfortunately, the previous findings shed some uncertainty in my mind as to whether the authors are actually only affecting H3K36me2 here, or whether some of their results – see also below – may be caused by other mechanisms.

Our response: We apologize that we did not discuss the apparent discrepancies pointed out by this reviewer in our original manuscript. However, as this reviewer states, the studies concern different cell types, and even different H3 isoforms, and therefore it is difficult to make a direct comparison. In our case, a knock-in approach was taken to express H3.3K36M at a physiological level, and our previous co-immunoprecipitation assay clearly showed that H3.3K36M interacts with NSD1 and NSD2 (H3K36me2 methyltransferases), not with SETD2 (H3K36me3 methyltransferase), consistent with the specific reduction of H3K36me2 (Abe, et al. 2020 and this study). The preferential inhibition of NSD2 by the K36M mutation was also reported by an independent group (Zhuang et al. 2018). Therefore, we have added such discussions in the results section (**page 6, paragraph 2, L95-100**).

6) The authors claim “Loss of H3K36me2 in oocytes causes embryonic lethality around implantation”, is weakened by the fact that they are expressing a mutant histone, with possibly other effects than inhibiting NSD enzymes. My first concern was that the tagged version of the histone may be detrimental. It took me some unnecessarily long searching to find out that the authors have already demonstrated that expression of the WT version of the tagged histone has no discernible phenotype – again prior work needs to be better indicated and explicitly referenced. Once I understood what is happening, I was impressed by the idea that half of the eggs should not continue to express K36M,

and their future lethality is only affected by the early depletion of H3K36me2. However, that is not quite true, since the mutant histones will persist in the zygote, and even at the ~100 cell stage should be present at ~ 1/50 of the initial levels (even if not detectable by staining).

Our response: We apologize for not having mentioned explicitly that HA-tagged wild-type H3.3 (without K36M) has no detrimental effect on mouse development and fertility. This was described in our previous report (Abe et al. 2021), and the information is now included in the results section (**page 6, paragraph 2, L103-104**). Regarding the second point, there should indeed be a carry-over of maternal H3.3K36M in zygotes and early embryos. As reviewer #1 raised the same point, please see our response to *Major comment 1* of that reviewer. In brief, the carry-over could also contribute to the phenotype, which should be resolved more clearly in future studies.

7) I was not convinced that the DM condition recapitulates DNMT3A KO methylation patterns, as the authors imply. It does result in low levels of methylation, which is interesting, but the correlation plot in figure 5c clearly shows two clusters, one that is more affected by DNMT3A than DM and one that is more affected by DM than DNMT3A. This should be explained. Similarly, I don't understand the premise behind figure 5f and the rationale of identifying regions that don't change in one condition and showing that they change little in the other condition. Biologically, what does this mean and why should this happen? Quite often, such extreme set of regions represent some artifact of the analysis, rather than having any biological relevance.

Our response: We never imply that *DM* oocytes recapitulate the methylation patterns of *Dnmt3a* KO oocytes and totally agree that there are regions showing some differences (**Fig. 5c,d**). We can think of many reasons for such differences: incomplete erasure of the histone marks, secondary changes in the related modification enzymes, different timing of the Cre driver expression (*Gdf9* versus *Zp3*), etc. Anyway, since most genomic regions (except the regions discussed later in new **Fig. 5f,g** and **Supplementary Fig. 9f,g**) showed greatly reduced methylation in both genotypes and the differences between the genotypes were small (Δ CG methylation < 10%), we have decided not to pursue this further. However, we now describe the presence of the differences explicitly in the results section (**page 12, paragraph 1, L206-207**). Regarding the second point, we apologize that the logic behind this analysis was not explained well in the original manuscript. What we wanted to say is that, when we focused on the rare genomic regions that were unaffected by *Dnmt3a* KO (=regions that are not de novo methylated by *Dnmt3a*), such regions were also unaffected by *DM*, further corroborating the link between H3K36me2/me3 and de novo methylation by *Dnmt3a*. To better present this finding, we have created new **Fig. 5f** and **Supplementary Fig. 9f** with an appropriate control. In addition, we have performed an additional analysis and found that the regions unaffected by neither *Dnmt3a* KO nor *DM* are already CG-methylated in non-growing oocytes (new **Fig. 5g** and **Supplementary Fig. 9g**), suggesting that they are not subject to de novo methylation in oocytes. Therefore, we have revised the corresponding portions of the results section (**page 12, paragraph 1, L213-219**) and the methods section (**page 21, paragraph 2, L373-376**).

8) The discussion is very terse and should be better used to position this paper within the existing literature, as well as to clarify some of the concerns mentioned above.

Our response: We appreciate the reviewer's suggestion. We have added a paragraph, mainly on the unique H3K36me2 pattern on the X chromosome, in the discussion section (**page 13, paragraph 2, L233-250**). In relation to this discussion, some details regarding the X chromosome are added in the result section as well (**page 5, paragraph 1, L74-78**). We believe that this addition has improved the quality of the discussion.

Addressing the above points will improve the manuscript, in my opinion. Overall, I think that the analysis is sound, and I appreciate the value of the work. However, the remaining concern will still be the moderate level of novelty for this level of publication.

Our response: We appreciate the helpful comments from this reviewer and hope that the manuscript has been satisfactorily improved.

REVIEWERS' COMMENTS

Reviewer #1 (Remarks to the Author):

The authors have addressed all of my concerns satisfactorily. One issue that had come to my mind subsequently to submitting my first review was the number of publications that had demonstrated that H3K36M impacts H3K9me3, which evidently is not the case in oocytes. However, it appears the other reviewer also raised this point, and the authors addressed it. In fairness, I will leave it to the other reviewer to assess the author's response.

Maxim Greenberg

Reviewer #2 (Remarks to the Author):

The authors have appropriately addressed my concerns. I appreciate the additional effort and recommend the manuscript for publication.

Jacek Majewski